analytical chemistry

deep eutectic solvent, emulsification liquid–liquid microextraction, phenoxy acid herbicides, high-performance liquid chromatography, dicamba, MCPA

**Author for correspondence:**
Mazidatulakmam Miskam
e-mail: mazidatul@usm.my

This article has been edited by the Royal Society of Chemistry, including the commissioning, peer review process and editorial aspects up to the point of acceptance.

# Deep eutectic solvent-based emulsification liquid–liquid microextraction for the analysis of phenoxy acid herbicides in paddy field water samples

Nur 'An Nisaa Mohamad Yusoff[1], Nurul Yani Rahim[1], Rania Edrees Adam Mohammad[1], Noorfatimah Yahaya[2] and Mazidatulakmam Miskam[1]

[1]School of Chemical Sciences, Universiti Sains Malaysia, 11800 USM Pulau Pinang, Malaysia
[2]Integrative Medicine Cluster, Advanced Medical and Dental Institute (AMDI), Universiti Sains Malaysia, 13200 Bertam, Kepala Batas, Pulau Pinang, Malaysia

NYR, 0000-0001-8139-2867; NY, 0000-0002-3079-7837;
MM, 0000-0001-9757-0883

An emulsification liquid–liquid microextraction (ELLME) method was successfully developed using phenolic-based deep eutectic solvent (DES) as an extraction solvent for the determination of phenoxy acid herbicides, 3,6-dichloro-2-methoxybenzoic acid (dicamba) and 2-methyl-4-chlorophenoxyacetic acid (MCPA) in environmental water samples. Five different phenolics-based DESs were successfully synthesized by using phenol (DES 1), 2-chlorophenol (DES 2), 3-chlorophenol (DES 3), 4-chlorophenol (DES 4) and 3,4-dichlorophenol (DES 6) as the hydrogen-bond donor (HBD) and choline chloride as the hydrogen-bond acceptor (HBA). The DESs were mixed at 1 : 2 ratio. A homogeneous solution (clear solution) was observed upon the completion of successful synthesis. The synthesized DESs were characterized by using Fourier transform infrared and nuclear magnetic resonance (NMR). Under optimum ELLME conditions (50 µl of DES 2 as extraction solvent; 100 µl of THF as emulsifier solvent; pH 2; extraction time 5 min), enrichment factor obtained for dicamba and MCPA were 43.1 and 59.7, respectively. The limit of detection and limit of quantification obtained for dicamba were 1.66 and 5.03 µg l$^{-1}$, respectively, meanwhile for MCPA were 1.69 and 5.12 µg l$^{-1}$, respectively. The developed ELLME-DES method was applied

on paddy field water samples, with extraction recoveries in the range of 79–91% for dicamba and 82–96% for MCPA.

## 1. Introduction

Phenoxy acid herbicides are one of the commonly used herbicides to control the growth of broadleaf weeds, especially in crop plantations such as rice, corn, wheat, fruit trees, forestry and also in domestic usage [1,2]. Despite the effectiveness of these herbicides, their extensive usage poses a serious threat to directed-seed planting systems and exert serious environmental effects on non-target organisms [3]. Phenoxy acid herbicides can easily enter surface or groundwater through natural drainage or infiltration due to their high polar and good solubility, thus contaminating water sources [2,4,5]. Besides, due to the potentially teratogenic and carcinogenic properties, either direct contact or indirect exposure will be harmful to humans. Therefore, by virtue of well-known toxicity and detrimental effect, there is thus great interest in the monitoring and determination of phenoxy acid herbicides, facilitated by sensitive, selective and reproducible analytical methods in the environmental samples.

Due to the low concentration of phenoxy acid herbicides in environmental water, sample preparation is a crucial step in analytical procedures to achieve a low limit of detection (LOD) and high accuracy. Liquid–liquid extraction (LLE) and solid phase extraction (SPE) are the most commonly used for phenoxy acid herbicide residues analysis [6–9]. However, LLE is time-consuming and requires large volumes of toxic organic solvents. SPE is less solvent consumption method compared to LLE but requires cartridge conditioning and elution with organic solvents. Because of these shortcomings, miniaturized LLE named liquid-phase microextraction (LPME) has been developed in the last decades [10–12]. LPME methods are more miniaturized, easy and economic as only a few microlitres of organic are needed to extract analyte from the aqueous samples. One of the most recent developments of LPME methods is emulsification liquid–liquid microextraction (ELLME). In ELLME, extraction solvent with high solubility and an emulsifier solvent is added to the sample solutions to extract the target analyte. A cloudy solution resulted from the formation of small droplets in the medium is achieved. After centrifugation and separation of organic and aqueous layers, the amount of analyte extracted to the extraction phase is determined using an appropriate separation technique. Of the plethora of microextraction techniques, ELLME has shown several benefits including time-saving, lower consumption of organic solvents and higher extraction efficiency [13].

With the introduction of green chemistry, the selection of proper solvent has been becoming a crucial task not only for effective extraction but also for the development of green analytical methods. Over the past decade, a new type of ionic liquid solvents known as deep eutectic solvents (DESs) has garnered interest due to their unique properties. DESs are commonly synthesized by using a combination of ammonium salts (e.g. choline chloride; ChCl) and hydrogen bond donors (HBD), such as urea, carboxylic acids or polyols, which are economical, biodegradable and non-toxic [14–17]. The formation of hydrogen bonding between the halide anion of ChCl and functional groups of a hydrogen-donor agent is responsible for the decrease in the freezing point of DESs in relation to the melting points of the individual components [18,19]. The ability to donate or accept electrons or protons is probably the main characteristic of DESs.

The study on DES is not only confined to its physical properties but has been expanded to be applied in the chemical process such as in producing biodiesel [20] and lubricants [21]. Due to its designable properties, the application of DES has been extended in the field of analytical chemistry as well [22,23]. In the field of separation science, ChCl-phenol were used as extraction solvent for the determination of BTEX and PAH from water sample [13], curcumin in food and herbal tea samples [24], and caffeine in beverages [25]. Khezeli and Daneshfar have reported using ChCl/urea-based DES as the desorption solvent in the dispersive micro-solid-phase extraction (d-mSPE) for determination of dopamine, epinephrine and norepinephrine in the biological sample [26]. On the other hand, Khezeli *et al*. also studied the potential of ChCl/ethylene glycol and ChCl/glycerol-based DES to serve as an extraction solvent for the determination of ferulic, caffeic and cinnamic acid from vegetable oils using ultrasonic-assisted liquid–liquid microextraction [27]. Garcia *et al*. explored the synthesized DES with ChCl and different types of HBD; sugars, alcohol, organic acids and urea to extract phenolic compounds from virgin olive oil [28].

Despite the advantages of DESs, to the best of our knowledge, no extraction procedures based on these solvents have been reported for the determination of phenoxy acid herbicides in paddy field water samples. Thus, the main purpose of this work is the utilization of DESs in the emulsification liquid–liquid microextraction method (ELLME-DES) to extract phenoxy acid herbicides in paddy field

water samples prior to high-performance liquid chromatography–ultraviolet detection (HPLC-UV). In our study, different types of DESs were synthesized based on different compositions of ChCl as hydrogen-bond acceptor (HBA) and phenolic derivatives (phenol, 2-chlorophenol, 3-chlorophenol, 4-chlorophenol, 2,3-dichlorophenol, 3,4-dichlorophenol and 2,3,4-trichlorophenol) as HBD. DESs obtained under different preparation conditions and ratios were characterized using Fourier transform infrared (FTIR) spectrometry and nuclear magnetic resonance (NMR). Their performance as an extraction solvent in ELLME was evaluated based on extraction performance and pre-concentration of dicamba and MCPA as model analytes. The developed method has the advantages of high extraction recoveries and environment-friendly. Finally, the method was successfully applied to simultaneously determine phenoxy acid herbicides in three paddy field sampling sites under optimal conditions. Our work provides insight to contribute to the urgent need for green chemistry for phenoxy acid herbicides residues in the paddy field water sample.

# 2. Material and methods

## 2.1. Chemicals and materials

Hydrochloric acid (37%), ortho-phosphoric acid (85%), acetone and acetonitrile with analytical grade were obtained from QRëC (Malaysia). Tetrahydrofuran with HPLC grade was purchased from Merck (Darmstadt, Germany). Chemical standards of 3,6-dichloro-2-methoxy benzoic acid (dicamba) and 2-methyl-4-chlorophenoxy acetic acid (MCPA) were purchased from Supelco (Darmstadt, Germany). Choline chloride (ChCl) was purchased from Sigma (Darmstadt, Germany) with a purity of greater than or equal to 98%. Phenol was purchased from R&M Chemicals (London, UK). 2-chlorophenol, 3-chlorophenol, 4-chlorophenol, 2,3-dichlorophenol, 3,4-dichlorophenol and 2,3,4-trichlorophenol were obtained from Merck (Darmstadt, Germany). HPLC grade methanol was purchased from Merck (Darmstadt, Germany).

## 2.2. Instrumentations

Quantification of dicamba and MCPA was carried out using HPLC system (JASCO, USA), equipped with JASCO PU 158- Pump and UV-1570 Intelligent UV detector operating at a 235 nm wavelength. The chromatographic separation was done on Agilent analytical column, Licrosorb $C_{18}$ (150 × 4.6 mm i.d., 5 μm particle size). The mobile phase compositions consisted of methanol and 0.02 M phosphoric acid (55 : 45, v/v), the flow rate was set at 1 ml min$^{-1}$ using isocratic elution. FTIR spectra of synthesized DES were recorded using a Perkin Elmer FT-NIR Spectrometer (4000–600 cm$^{-1}$, resolution 4 cm$^{-1}$, 16 scans). The molecular structure of DESs was further confirmed based on proton nuclear magnetic resonance ($^1$H-NMR 400 MHz, DMSO-d6) using NMR Bruker Advance.

## 2.3. Synthesis of deep eutectic solvents

ChCl as HBA was mixed with phenol as HBD with 1 : 2 ratio in a 10 ml screw cap glass tube. The mixture was placed in a water bath at 80°C for 10 min then vortexed for 2 min. The step was repeated by placing the mixture in a water bath at 80°C for 20 min. The preparation of DES 1 (ChCl-phenol) was completed after the mixture was vortexed again for 2 min. A homogeneous solution (clear solution) was observed upon the completion of successful synthesis. The procedure was repeated to synthesize DES 2 (ChCl-2-chlorophenol), DES 3 (ChCl-3-chlorophenol), DES 4 (ChCl-4-chlorophenol), DES 5 (ChCl-2,3-dichlorophenol), DES 6 (ChCl-3,4-dichlorophenol) and DES 7(ChCl-2,3,4-trichlorophenol).

## 2.4. Emulsification liquid–liquid microextraction procedure

A 1.5 ml sample of ultrapure water containing 0.1 mg l$^{-1}$ of dicamba and MCPA of pH 2 was transferred into a centrifuge tube. A hundred millilitres of DES (extraction solvent) and THF (emulsifier solvent) were added to the sample. The tube was placed on the vortex mixer at 2500 r.p.m. for 5 min. The solution was cloudy after vortexing which was caused by the aggregation of DES droplets from the aqueous phase. The targeted analytes were simultaneously extracted to the DES phase during this process.

The solution was then centrifuged at 3400 r.p.m. for 10 min to separate the DES phase from the aqueous phase. A stock solution of individual pesticides (1000 mg l$^{-1}$) was prepared in methanol.

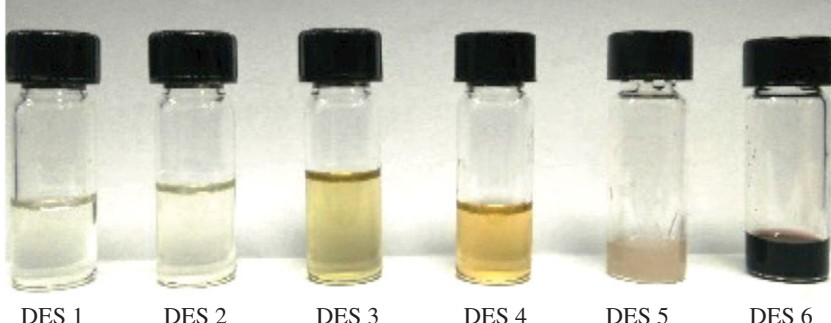

**Figure 1.** DES synthesized from different type of phenol derivatives as HBD; DES 1 (phenol); DES 2 (2-chlorophenol); DES 3 (3-chlorophenol); DES 4 (4-chlorophenol); DES 5 (2,3-dichlorophenol) and DES 6 (3,4-dichlorophenol).

A working solution containing dicamba and MCPA (each 100 mg l$^{-1}$) was prepared in methanol. Calibration standard solution (0.1–10 mg l$^{-1}$) were prepared in ultra-pure water.

## 2.5. Pre-treatment of the collected paddy field water samples

The water samples were collected from three different paddy fields across the state of Kedah and Pulau Pinang, Malaysia and were stored at a temperature less than or equal to 4°C until the commencement of analysis. The pH of samples was adjusted to 2 with 0.1 M HCl solution and filtered with 0.22 µm nylon membrane filter for the extraction procedure.

# 3. Results

## 3.1. Synthesis of DESs

The nature of DESs may have a remarkable effect on the successfulness of the synthesis of DESs and the extraction efficiency of the analytes. Based on the naked eye observation, the DESs were formed by obtaining a clear solution when HBA and HBD were mixed. As depicted in figure 1, DES 1, DES 2, DES 3, DES 4, DES 5 and DES 6 were successfully synthesized.

With chlorophenol derivatives, the successfulness of the DES synthesis may be subjected to the different positions of chlorine atoms as HBD. The presence of a chlorine atom in the phenolic compounds reduced the electron density of the hydroxyl group thus increasing its acidity [29]. This will further enhance the ability to form hydrogen bonds with choline chloride. The position and number of chlorine atoms will greatly influence the stability or the formation of this hydrogen bond [30]. The formation of DES 7 was unsuccessful as a clear solution was not obtained until the end of the synthesis. This may be due to the additional chlorine atom present in the para-position resulting in a non-favourable formation of hydrogen bond with ChCl as the ring was further deactivated [29].

However, after further observation, DES 5 became unstable where a formation of clear liquid was obtained upon completion of the synthesis process, turned into a solid form after a few days at room temperature. The presence of chlorine atom in the ortho-position might contribute to a formation of an intramolecular hydrogen bond with the hydroxyl group resulting in a weak hydrogen bond interaction between the phenolic derivative and choline chloride [29]. The interactions were further weakened with the presence of choline atom in meta-position which further deactivates the ring thus contributing to the instability of DES structure [30]. The molar ratio of HBA : HBD is kept constant at 1 : 2; it was proposed as the optimum ratio for the DES synthesized from ChCl/4-chlorophenol consistent with previously reported study [31].

### 3.1.1. FTIR

The main force for the formation of DES is hydrogen binding between halide anion in ChCl and hydroxyl group in phenol derivatives. Based on the IR spectra (figure 2a), the broad band indicating a stretching vibration for the –OH group in each DES have shifted from the initial value for the phenolic derivatives. The shifting of –OH groups is summarized in table 1. The shifting might be due to the transfer of electrons cloud from oxygen atom to form hydrogen bond and is an indication of a formation of

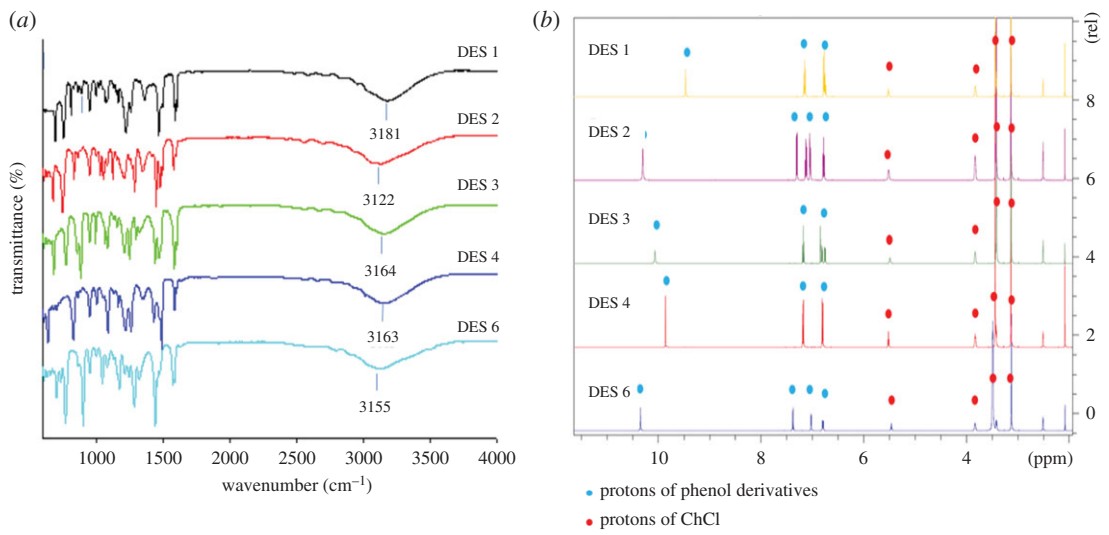

**Figure 2.** (*a*) FT-IR spectrum and (*b*) [1]H-NMR of DES 1, DES 2, DES 3, DES 4, DES 6.

**Table 1.** FTIR wavenumber, cm[−1] of hydroxyl (−OH) group in HBD and DES.

| DES (HBA: HBD) | wavenumber of −OH group (cm[−1]) | |
| --- | --- | --- |
| | HBD | DES |
| DES 1 (ChCl: phenol) | 3313 | 3181 |
| DES 2 (ChCl: 2-chlorophenol) | 3519 | 3122 |
| DES 3 (ChCl: 3- chlorophenol) | 3329 | 3164 |
| DES 4 (ChCl: 4-chlorophenol) | 3328 | 3163 |
| DES 6 (ChCl: 3,4-dichlorophenol) | 3244 | 3155 |

stable DES [13,32]. Besides, the affinity of hydrogen binding to absorb oxygen atom cloud point and slightly decrease its force constant. Therefore, the shift of the O–H vibration bond suggests the existence of hydrogen bonding and formation of DES.

### 3.1.2. [1]H-NMR

The spectra of DESs were deduced from [1]H-NMR with the tabulated assignment of functional groups in figure 2*b* were carried out to confirm the purities of the synthesized DESs. As summarized in table 2, the spectra of DESs consisted of singlet peak for the OH ($\delta$ = 9.46–10.3 ppm) and multiplet peaks for phenyl ($\delta$ = 6.62–7.55 ppm), attributed to the phenol and its derivatives (2-chlorophenol, 3-chlorophenol, 4-chlorophenol and 3,4-dichlorophenol). The [1]H spectra of DESs show the peaks of ChCl through the existence of singlet peak for the OH ($\delta$ = 5.52 and 5.46, 1H), multiplet peaks for $CH_2$ ($\delta$ = 3.82–3.84, 2H; $\delta$ = 3.38–3.42, 2H) and multiplet peaks for $(CH_3)_3$ ($\delta$ = 3.12 and 3.13, 9H). Besides, the downfield shift for the hydroxy peak of phenolic derivatives is due to the hydrogen bonding formation with ChCl. Based on the result depicted, all the peaks have been attributed DESs components, and no extra peaks were found in the [1]H spectra indicating that no side reactions occurred and these DESs were pure.

## 3.2. Optimization parameters for ELLME

### 3.2.1. Selection of DES as the extraction solvent

The selection of DESs as a convenient extraction solvent is based on the electrostatic and π–π interactions with target analytes. Accordingly, the efficiency of DES 1, DES 2, DES 3, DES 4 and DES 6 were tested as an extraction solvent for ELLME. Due to interfering background during the analysis with HPLC-UV,

**Table 2.** [1]H data for DESs.

| type of DESs | chemical shift, $\delta$ (ppm) |
|---|---|
| DES 1 | ChCl: $\delta = 3.13$ (s, 9H, **CH$_3$**), 3.42 (m, 2H, **CH$_2$**), 3.82 (m, 2H, **CH$_2$**), 5.52 (s, 1H, **OH**); phenol: $\delta = 6.76$ (m, 3H, **CH**), 7.15 (t, 2H, **CH**), 9.46 (s, 1H, **OH**) |
| DES 2 | ChCl: $\delta = 3.13$ (s, 9H, **CH$_3$**), 3.38 (m, 2H, **CH$_2$**), 3.83 (m, 2H, **CH$_2$**), 5.52 (s, 1H, **OH**); 2-chlorophenol: $\delta = 6.78$ (t, 1H, **CH**), 7.04 (d, 1H, **CH**), 7.12 (t, 1H, **CH**), 7.30 (d, 1H, **CH**), 10.3 (s, 1H, **OH**) |
| DES 3 | ChCl: $\delta$ 3.13 (s, 9H, **CH$_3$**), 3.42 (m, 2H, **CH$_2$**), 3.82 (m, 2H, **CH$_2$**), 5.52 (s, 1H, **OH**); 3-chlorophenol: $\delta = 6.75$ (d, 1H, **CH**), 6.81 (d, 1H, **CH**), 6.84 (s, 1H, **CH**), 7.17 (t, 1H, **CH**), 10.0 (s, 1H, **OH**) |
| DES 4 | ChCl: $\delta = 3.13$ (s, 9H, **CH$_3$**), $\delta$ 3.42 (m, 2H, **CH$_2$**), $\delta$ 3.82 (m, 2H, **CH$_2$**), $\delta$ 5.52 (s, 1H, **OH**); 4-chlorophenol: $\delta = 6.76$ (d, 2H, **CH**), 7.01 (d, 2H, **CH**), 10.3 (s, 1H, **OH**) |
| DES 6 | ChCl: $\delta$ 3.12 (s, 9H, **CH$_3$**), $\delta$ 3.42 (m, 2H, **CH$_2$**), $\delta$ 3.84 (m, 2H, **CH$_2$**), $\delta$ 5.46 (s, 1H, **OH**); 3,4-dichlorophenol: $\delta = 6.62$ (d, 1H, **CH**), 7.10 (s, 1H, **CH**), 7.55 (d, 1H, **CH**)10.3 (s, 1H, **OH**) |

the enrichment factor (EF) for DES 3 cannot be obtained because the peaks of dicamba and MCPA were not resolved due to the background interference from the DES itself. Meanwhile, the liquid extract of DES 5 crystallized upon drying under a stream of nitrogen gas possibly due to a higher freezing point property [14,30].

The performance of DES 1, DES 2 and DES 4 was evaluated based on EF values as shown in figure 3. In DES 1, the peak for dicamba was not resolved due to the overlapping with the DES peak. This is due to the high adsorption of DES 1 in the ultraviolet (UV) region which interferes with dicamba detection when HPLC-UV was used [33]. Based on the results, the highest EF for MCPA and dicamba were obtained with DES 2 and DES 4, respectively. The ability of chlorine atoms in the ortho- position in DES 2 to form an intramolecular hydrogen bond with the hydroxyl group contributes to a weaker hydrogen bond between HBA and HBD, resulting in a lower surface tension for DES 2. The reduction in the surface tension produces a smaller droplet which will subsequently increase the rate of transfer of analyte from the aqueous phase [22,24,34]. Therefore, as DES 2 provided a higher EF for phenoxy acid herbicides, thus DES 2 was chosen for the subsequent optimization.

### 3.2.2. Volume of DES

The volume of DES was studied from 25 to 200 µl and is presented in figure 4. The optimum volume of DES 2 was 50 µl for extraction of dicamba and MCPA. However, the EF decreased when the volume of DES was increased to 100, 150 and 200 µl. This decrease might be due to a low ratio between DES and emulsifier solvent, resulting in a lower droplet formation [35,36]. Thus, 50 µl was chosen for the subsequent analysis.

### 3.2.3. Type of emulsifier solvent

In the ELLME-DES method self-aggregation and separation of DES molecules in aqueous phase occurred by adding aprotic solvent. Theoretically, water molecules will interact favourably towards the emulsifier solvent instead of the DES molecule thus DES tends to self-aggregate in the form of fine droplets. This process increases the contact area between DES and the aqueous phase which results in an enhancement in the mass transfer of the analytes from the aqueous phase to the DES phase [13,37].

Different types of aprotic solvent as an emulsifier solvent such as ACN ($\varepsilon = 36$), acetone ($\varepsilon = 20.7$), ethyl acetate ($\varepsilon = 6$), THF ($\varepsilon = 7.6$) and DCM ($\varepsilon = 8.9$) were used. The highest EF for the extraction of dicamba and MCPA using THF as an emulsifier is shown in figure 5. The phase separation did not occur while using ACN and acetone as emulsifier solvents due to the high polarity of both solvents resulting in a preferred interaction of DES with water. As for DCM, it possesses the highest dielectric constant value in comparison with ethyl acetate and THF which theoretically gives out the highest polarity. DCM is only soluble in water to a certain extent due to the absence of oxygen atom in its structure resulting in a low ability to interact with water, thus lowering the self-aggregation rate of DES. Thus, further optimization was proceeded using THF as the emulsifier solvent.

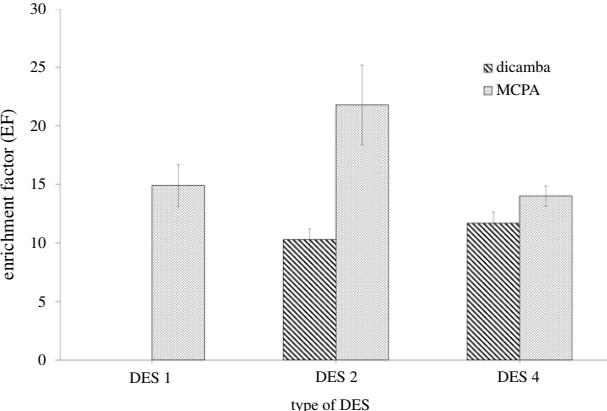

**Figure 3.** Effect of type of DESs on the enrichment factor. ELLME conditions: DES volume, 100 µl; emulsifier solvent, 100 µl THF; pH of sample, 3; extraction time, 15 min; phenoxy acid herbicides concentration, 100 µg l$^{-1}$.

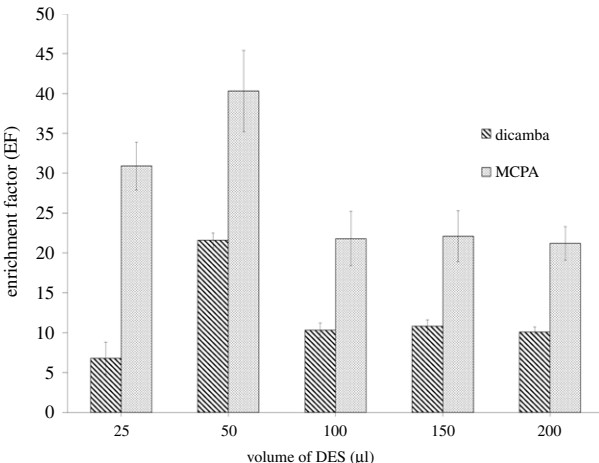

**Figure 4.** Effect of volume of DES on the enrichment factor. ELLME conditions: DES type, DES 2; emulsifier solvent, 100 µl THF; pH of sample, 3; extraction time, 15 min; phenoxy acid herbicides concentration, 100 µg l$^{-1}$.

### 3.2.4. Volume of emulsifier solvent

The volume of the emulsifier (THF) was studied from 25 to 200 µl. Based on figure 6, the optimum volume of emulsifier to extract dicamba and MCPA was 100 µl. The solubility of analytes in the aqueous phase started to increase with an increase in the volume of THF to 200 µl, causing a decrease in the EF value. Meanwhile, the low value of EF while using THF less than 100 µl may be due to the lack of DES aggregation.

### 3.2.5. pH of sample

The pH of an aqueous sample contributes to form of the analyte which subsequently affects the extraction efficiency. As depicted in figure 7, the pH of samples was varied from pH 2 to 6. The p$K_a$ of dicamba and MCPA is 1.97 and 3.07, respectively. At pH 2, the ionization of dicamba was partially suppressed while MCPA was fully suppressed where the herbicides were preferable to be extracted into the DES [38]. However, the EF started to decrease when the pH was increased. This may be due to the ionization of the herbicides which reduced the extraction efficiency. Thus, pH 2 was chosen as the optimum pH for extraction of dicamba and MCPA.

### 3.2.6. Extraction time

The dispersion of extraction solvent into the aqueous phase was assisted by vortexing the solution at a certain speed and time until an equilibrium is reached. Therefore, various vortexing times were applied

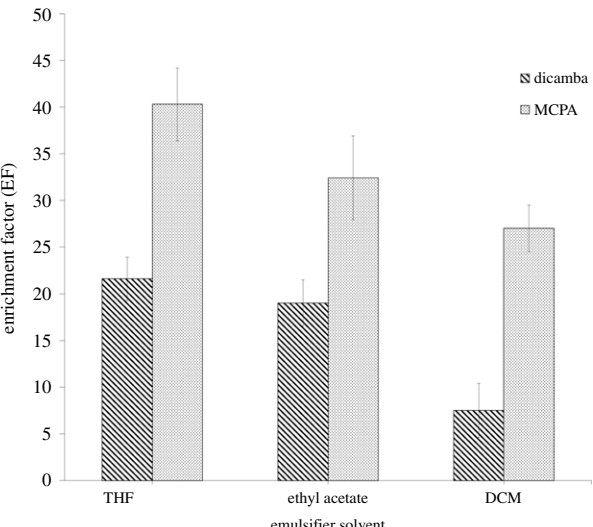

**Figure 5.** Effect of type of emulsifier solvent on the enrichment factor. ELLME conditions: DES type, DES 2; DES volume, 50 µl; emulsifier solvent volume, 100 µl; pH of sample, 3; extraction time, 15 min; phenoxy acid herbicides concentration, 100 µg l$^{-1}$.

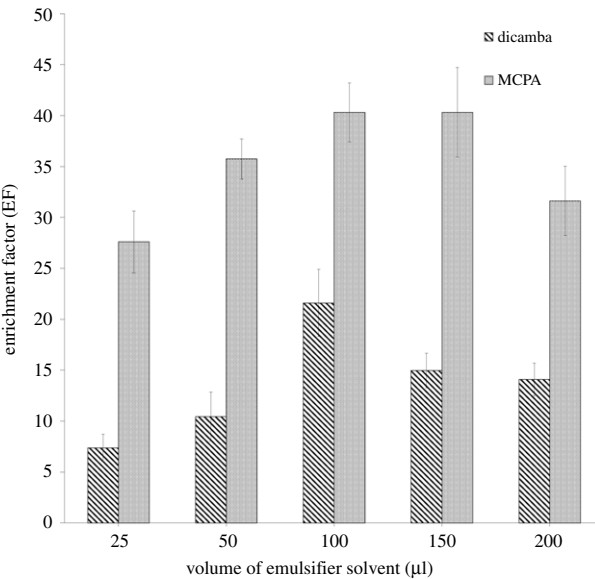

**Figure 6.** Effect of volume of emulsifier solvent on the enrichment factor. ELLME conditions: DES type, DES 2; DES volume, 50 µl, emulsifier solvent, THF; pH of sample, 3; extraction time, 15 min; phenoxy acid herbicides concentration, 100 µg l$^{-1}$.

at a constant speed. As shown in figure 8, the lowest EF was obtained when the samples are not vortexed due to inefficient mass transfer. When the vortex was applied for 5 min, the EF started to increase as the mass transfer increased. However, the EF was slightly decreasing with the prolonged vortex time due to back-extraction. Thus, 5 min was employed for the subsequent analysis.

## 3.3. Analytical performance of ELLME-DES

The analytical performance of the proposed method was evaluated based on some of the analytical parameters that include the linear ranges, correlation coefficient, LOD, LOQ, EF and repeatability expressed in terms of relative standard deviation (RSD). All of these parameters were studied under the optimum experimental condition. The summary of the results is tabulated in table 3. The linear range for the calibration graph was 5–100 µg l$^{-1}$ with a correlation coefficient equal to or higher than 0.999 for dicamba and MCPA. The LOD for dicamba and MCPA was 1.66 and 1.69 µg l$^{-1}$, respectively. Meanwhile, the LOQ for dicamba and MCPA was 5.03 and 5.12 µg l$^{-1}$, respectively. The

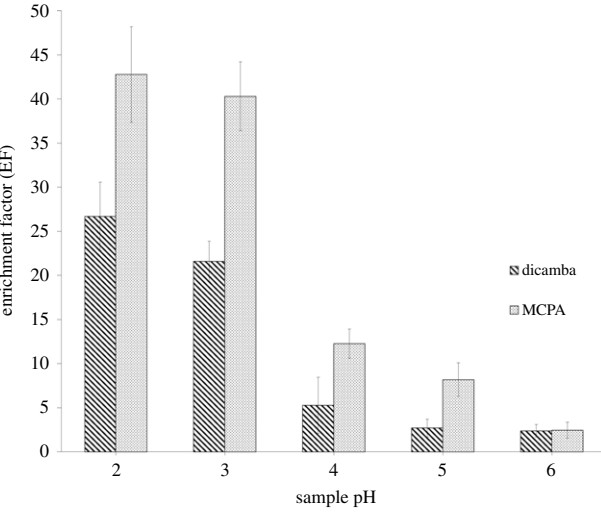

**Figure 7.** Effect of pH of sample on the enrichment factor. ELLME conditions: DES type, DES 2; DES volume, 50 µl; emulsifier solvent, THF, 100 µl; extraction time, 15 min; phenoxy acid herbicides concentration, 100 µg l$^{-1}$.

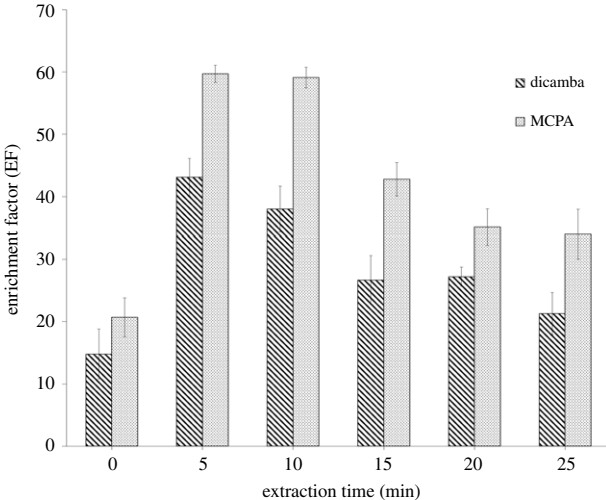

**Figure 8.** Effect of extraction time on the enrichment factor. ELLME conditions: DES type, DES 2; DES volume, 50 µl, emulsifier solvent, THF, 100 µl; sample pH, 2; phenoxy acid herbicides concentration, 100 µg l$^{-1}$.

**Table 3.** Method validation of ELLME-DES for the determination of phenoxy acid herbicides.

| | | | | | | RSD (%) | |
|---|---|---|---|---|---|---|---|
| analytes | concentration (µg l$^{-1}$) | $R^2$ | LOD (µg l$^{-1}$) | LOQ (µg l$^{-1}$) | EF (%) | intra-day (n = 5) | inter-day (n = 3) |
| dicamba | 5–100 | 0.999 | 1.66 | 5.03 | 43.1 | 2.9 | 4.6 |
| MCPA | 5–100 | 0.999 | 1.69 | 5.12 | 59.7 | 4.5 | 5.1 |

EF obtained for dicamba and MCPA was 43.1 and 59.1, respectively. The repeatability was evaluated by extracting a water sample containing 50 µg l$^{-1}$ of dicamba and MCPA. The RSD for intra-day (n = 5) for dicamba and MCPA was 2.9% and 4.5%, respectively. Meanwhile, the RSD for inter-day (n = 3) was 4.6% and 5.1% for dicamba and MCPA, respectively.

**Table 4.** Recovery of dicamba and MCPA from paddy field water samples ($n = 3$). N.D., not detected.

| sample no. | blank (µg l$^{-1}$) | | mean recovery for spiked sample, % ± s.d. | |
| --- | --- | --- | --- | --- |
| | dicamba | MCPA | dicamba | MCPA |
| sample 1 | N.D. | N.D. | 82 ± 6.4 | 82 ± 2.5 |
| sample 2 | N.D. | N.D. | 79 ± 6.5 | 92 ± 2.3 |
| sample 3 | N.D. | N.D. | 91 ± 4.3 | 96 ± 7.5 |

## 3.4. Analysis of real samples

The applicability of the proposed method was further demonstrated by using the optimum condition by analysing water samples collected from paddy fields at different locations across the state of Pulau Pinang and Kedah, Malaysia. No phenoxy acid herbicides are detected in the blank sample.

The spiked samples at a concentration of 50 µg l$^{-1}$ were analysed to calculate the recoveries of the herbicides. The experiments based on independently prepared samples were performed in triplicate. The analytical results are listed in table 4. As seen, the developed method provided recoveries in the range of 79–91% for dicamba and 82–96% for MCPA with s.d.s lower than 7.5%. The recoveries of the remaining target analytes were satisfactory, indicating no matrix interference.

## 4. Conclusion

In this work, DES 1, DES 2, DES 3, DES 4 and DES 6 were successfully synthesized using choline chloride and phenol derivatives such as phenol, 2-chlorophenol, 3-chlorophenol, 4-chlorophenol and 3,4-dichlorophenol, respectively. The synthesis of DES, however, was unsuccessful using 2,3-dichlorophenol and 2,3,4-trichlorophenol as HBD. This study revealed that the number of chlorine atom and the presence of chlorine atom located next to the hydroxyl group of phenol derivatives would affect the stability of DES.

The performance of DES as an extraction solvent was evaluated in the newly developed ELLME method for the determination of dicamba and MCPA in the water samples. The highest EF for the extraction of dicamba and MCPA were obtained by using DES 2. Under optimum ELLME condition, (50 µl of DES 2 as extraction solvent; 100 µl of THF as emulsifier solvent; pH 2; extraction time; 5 min) a wide linear range, low LOD and LOQ, high EF and good repeatability in intra- and inter-day validation were obtained which approved that method is a suitable green alternative for sample preparation in terms of performance and speed.

Data accessibility. Data are available from the Dryad Digital Repository: https://doi.org/10.5061/dryad.v15dv41v3 [39].
Authors' contributions. N.Y.R., N.Y., R.E.A.M. and M.M. participated in the design of the study; N.A.N.M.Y. and R.E.A.M carried out most of the laboratory work, accomplished the data analysis and drafted the manuscript; N.Y.R, N.Y. and M.M conceived the study, coordinated the study and helped draft the manuscript. All authors gave final approval for publication.
Competing interests. The authors declare that they have no competing interests.
Funding. This work was supported by financial funding from Universiti Sains Malaysia Research Grants (Short-term Grant 304/PKIMIA/6315106) and the Ministry of Higher Education (FRGS-203/PKIMIA/6711641, FRGS-203/PKIMIA/6711924).
Acknowledgements. The authors thank the editors and anonymous reviewers for their helpful suggestions for this manuscript.

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
