## [Peer Review File · Royal Society Open Science]

Review History

RSOS-202061.R0 (Original submission)

Review form: Reviewer 1

Is the manuscript scientifically sound in its present form?

Yes

Are the interpretations and conclusions justified by the results?

No

Is the language acceptable?

Yes

Do you have any ethical concerns with this paper?

No

Have you any concerns about statistical analyses in this paper?

No

Recommendation?

Major revision is needed (please make suggestions in comments)

Comments to the Author(s)

1. In the first use of some terms, their full names must be written before synonyms are used. For example, MCPA,
2. Mean recoveries around 27 ± 3.5 and 34 ± 3.2 are not acceptable for analytical methods. How do authors explained this situation.
3. What is the last volume of the extraction phase before HPLC analysis?
4. How author did reach to about 50-60 enrichment factor?
5. All units should be given same for example mg L-1 or mg/L
6. Why did author use enrichment factor instead of recovery % for the optimization stage?

Review form: Reviewer 2

Is the manuscript scientifically sound in its present form?

Yes

Are the interpretations and conclusions justified by the results?

Yes

Is the language acceptable?

Yes

Do you have any ethical concerns with this paper?

No

Have you any concerns about statistical analyses in this paper?

No

Recommendation?

Major revision is needed (please make suggestions in comments)

Comments to the Author(s)

In this paper, an emulsification liquid-liquid microextraction (ELLME) method was developed by using phenolic deep eutectic solvent (DES) as extraction solvent for the determination of phenoxy acid herbicides, dicamba and MCPA in environmental water samples. But, there were some questions.

I have the following questions.

1. The manuscript should be revised carefully at least twice, there were too many errors should be revised.
2. The spectrogram of FTIR and ¹H-NMR should be list in the figures.
3. In section 3.3, is there any solvent was used to dissolve the substrate in the synthesis?
4. The major problem in this manuscript is the effect of matrix was not investigated. In section 3.4, the calibration standard solution was prepared in water. For the reliability of the method, the standard solution should be proceeding with the same process and dissolve in the same solvent with samples.

Review form: Reviewer 3 (Tasneem Kazi)

Is the manuscript scientifically sound in its present form?

Yes

Are the interpretations and conclusions justified by the results?

Yes

Is the language acceptable?

No

Do you have any ethical concerns with this paper?

No

Have you any concerns about statistical analyses in this paper?

No

Recommendation?

Major revision is needed (please make suggestions in comments)

Comments to the Author(s)

In this work, The developed method has the advantages of high extraction recoveries and environmental friendly. Whereas the method was successfully applied to simultaneously determine phenoxy acid herbicides in 5 paddy field sampling sites under the optimal conditions. The work is provides insight to contribute the urgent need of green chemistry for phenoxy acid herbicides residues in paddy field water sample.

The title must be rephrase, mention only copper whereas authors added cadmium and lead

- Large number of grammatical and typing errors are present
- Don't repeat same word in sentence for many time
- The experimental section must be carefully checked

Sampling must be improved

- Mention results according to three significant rule

Decision letter (RSOS-202061.R0)

This year has been very difficult for everyone, and we want to take the opportunity to thank you for your continued support in 2020.

The Royal Society Open Science editorial office will be closed from the evening of Friday 18 December 2020 until Monday 4 January 2021. We will not be responding during this time. If you have received a deadline within this time period, please contact us as soon as possible to allow us to extend the deadline. If you receive any automated messages during this time asking you to meet a deadline, we offer apologies and invite you to respond after the festive period or during normal working hours.

With our best for a peaceful festive period and New Year, and we look forward to working with you in 2021.

Dear Dr Miskam:

Title: Deep eutectic solvent-based emulsification liquid-liquid microextraction for the analysis of phenoxy acid herbicides in paddy field water samples
Manuscript ID: RSOS-202061

The editor assigned to your manuscript has now received comments from reviewers. We would like you to revise your paper in accordance with the referee and Subject Editor suggestions which can be found below (not including confidential reports to the Editor). Please note this decision does not guarantee eventual acceptance.

Please submit your revised paper before 15-Jan-2021. Please note that the revision deadline will expire at 00.00am on this date. If we do not hear from you within this time then it will be assumed that the paper has been withdrawn. In exceptional circumstances, extensions may be possible if agreed with the Editorial Office in advance. We do not allow multiple rounds of revision so we urge you to make every effort to fully address all of the comments at this stage. If deemed necessary by the Editors, your manuscript will be sent back to one or more of the original reviewers for assessment. If the original reviewers are not available we may invite new reviewers.

RSC Associate Editor:
Comments to the Author:
(There are no comments.)

RSC Subject Editor:
Comments to the Author:
(There are no comments.)

Reviewers' Comments to Author:
Reviewer: 1

Comments to the Author(s)

1. In the first use of some terms, their full names must be written before synonyms are used. For example, MCPA,
2. Mean recoveries around 27 ± 3.5 and 34 ± 3.2 are not acceptable for analytical methods. How do authors explained this situation.
3. What is the last volume of the extraction phase before HPLC analysis?
4. How author did reach to about 50-60 enrichment factor?
5. All units should be given same for example mg L-1 or mg/L
6. Why did author use enrichment factor instead of recovery % for the optimization stage?

Reviewer: 2

Comments to the Author(s)

In this paper, an emulsification liquid-liquid microextraction (ELLME) method was developed by using phenolic deep eutectic solvent (DES) as extraction solvent for the determination of phenoxy acid herbicides, dicamba and MCPA in environmental water samples. But, there were some questions.

I have the following questions.

1. The manuscript should be revised carefully at least twice, there were too many errors should be revised.
2. The spectrogram of FTIR and $^1\text{H-NMR}$ should be list in the figures.
3. In section 3.3, is there any solvent was used to dissolve the substrate in the synthesis?
4. The major problem in this manuscript is the effect of matrix was not investigated. In section 3.4, the calibration standard solution was prepared in water. For the reliability of the method, the standard solution should be proceeding with the same process and dissolve in the same solvent with samples.

Author's Response to Decision Letter for (RSOS-202061.R0)

See Appendix A.

Decision letter (RSOS-202061.R1)

Dear Dr Miskam:

Title: Deep eutectic solvent-based emulsification liquid-liquid microextraction for the analysis of phenoxy acid herbicides in paddy field water samples
Manuscript ID: RSOS-202061.R1

It is a pleasure to accept your manuscript in its current form for publication in Royal Society Open Science. The chemistry content of Royal Society Open Science is published in collaboration with the Royal Society of Chemistry.

RSC Associate Editor:
Comments to the Author:
(There are no comments.)

RSC Subject Editor:
Comments to the Author:
(There are no comments.)

Reviewer(s)' Comments to Author:

Appendix A

Date: 8th February 2021

Manuscript ID: RSOS-202061

Title: "Deep eutectic solvent-based emulsification liquid-liquid microextraction for the analysis of phenoxy acid herbicides in paddy field water samples"

Dear Dr Laura Smith,

Editor,

Royal Society Open Science,

Thank you very much for organizing the review of our manuscript (RSOS-202061). We would like to thank the reviewers for their careful reading and thoughtful comments of our manuscript. We have considered all the comments given by the reviewers and revised the manuscript accordingly (The changes are highlighted).

We believe that we have addressed all the points raised by the reviewers. We hope that the manuscript is now ready for publication in Royal Society Open Science.

Yours sincerely,

Dr. Mazidatulakmam Miskam

School of Chemical Sciences,

Universiti Sains Malaysia,

11800 USM Minden,

Pulau Pinang, Malaysia.

mazidatul@usm.my